

# CRYAB predicts clinical prognosis and is associated with immunocyte infiltration in colorectal cancer

Junsheng Deng[1],[*], Xiaoli Chen[1],[*], Ting Zhan[1], Mengge Chen[1], Xisheng Yan[2] and Xiaodong Huang[1]

[1] Gastroenterology, Tongren Hospital of Wuhan University, Wuhan, Hubei, China
[2] Tongren Hospital of Wuhan Unversity, Wuhan, Hubei, China
[*] These authors contributed equally to this work.

## ABSTRACT

**Background**. αB-Crystallin (CRYAB) is differentially expressed in various tumors. However, the correlation between CRYAB and immune cell infiltration in colorectal cancer (CRC) remains unclear.

**Materials & Methods**. Kaplan–Meier survival curves in The Cancer Genome Atlas (TCGA) were used to evaluate the relationship between CRYAB expression and both overall survival and progression-free survival. The relationships between CRYAB expression and infiltrating immune cells and their corresponding gene marker sets were examined using the TIMER database.

**Results**. The expression of CRYAB was lower in CRC tumor tissues than in normal tissues ($P < 0.05$). High CRYAB gene expression and high levels of CRYAB gene methylation were correlated with high-grade malignant tumors and more advanced tumor, nodes and metastasis (TNM) cancer stages. In addition, in colorectal cancer, there was a positive correlation between CRYAB expression and immune infiltrating cells including neutrophils, macrophages, CD8 + T cells, and CD4 + T cells, as well as immune-related genes including CD2, CD3D, and CD3E. Methylation sites such as cg13084335, cg15545878, cg13210534, and cg15318568 were positively correlated with low expression of CRYAB.

**Conclusion**. Because CRYAB likely plays an important role in immune cell infiltration, it may be a potential tumor-suppressor gene in CRC and a potential novel therapeutic target and predictive biomarker for colorectal cancer (CRC).

Corresponding author
Xiaodong Huang,
13297056720@163.com

## INTRODUCTION

Colorectal cancer (CRC) is one of the three most common cancers in the United States (*Lurje et al., 2008*; *Lee et al., 2018*) (after lung cancer, prostate cancer in men, and breast cancer in women) (*Altun et al., 2013*). According to estimates by the International Agency for Research on Cancer, in 2020, 1.9 million people worldwide were diagnosed (6% of all cancer diagnoses) and 935,000 people died of CRC (5.8% of all cancer-related deaths) (*Sung et al., 2021*). Approximately 28% of CRCs occur in the rectum, with 22% of these
cases of CRC in the rectum involving the distal colorectal regions and 41% involving the proximal colorectal regions (*Cheng et al., 2011*). Up to 50% of CRC patients show metastatic disease at the time of diagnosis (*Calon et al., 2012*). Despite advances in screening and treatment options (*Jeon et al., 2018*), the 5-year survival rate of patients with metastatic disease is < 10% (*Brenner, Kloor & Pox, 2014*; *De Stefano & Carlomagno, 2014*; *Suman et al., 2016*). Biomarkers to help diagnose CRC early have yet to be identified, but personalized treatment strategies can improve the prognosis of patients with CRC (*Yiu & Yiu, 2016*). Increasing evidence supports the idea that malignant tumors depend on tumor cells and the tumor microenvironment, including extracellular matrix (ECM) molecules, inflammatory mediators, and immune cells (*Hanahan & Coussens, 2012*). Tumor-infiltrating immune cells are valuable for diagnosing cancer and identifying cancer progression and prognosis (*Mlecnik et al., 2018*; *Kang et al., 2019*; *Zhou et al., 2019*). The immune factors and immune cells that constitute the tumor immune microenvironment play a significant part in the progression and occurrence of anti-tumor immunity (*Berraondo et al., 2016*; *Chen et al., 2017*). Investigating the characteristics of tumor immune infiltration has been valuable for the treatment, evaluation, and diagnosis of many cancers (*Peng et al., 2019*; *Zhang et al., 2019a*; *Zhang et al., 2019b Wei et al., 2020*), and encouraging advances have been made in cancer immunotherapy in terms of treatment efficacy and long-term patient survival (*Vonderheide, Domchek & Clark, 2017*). Therefore, studying the tumor immune microenvironment of CRC is critical.

αB-Crystallin (CRYAB) has a C-terminal domain, an N-terminal domain, and a central domain (*Rajagopal et al., 2015*). CRYAB may induce epithelial-mesenchymal transition (EMT) in CRC by activating the ERK signaling pathway, and can be an underlying cancer biomarker used in the prognosis and diagnosis of CRC (*Li et al., 2017a*; *Li et al., 2017b*). In the nervous system, CRYAB plays a neuroprotective role in neurodegenerative diseases such as familial amyloidotic polyneuropathy, which is characterized by CRYAB overexpression (*Magalhaes, Santos & Saraiva, 2010*). However, no correlation has been found between CRYAB and immune cell infiltration. *Shi et al. (2014)* showed that CRYAB is correlated with poor prognosis in CRC and promotes the invasion and metastasis of CRC via epithelial-mesenchymal transition (EMT) (*Shi et al., 2017*) However, the role of an underlying marker for the prognosis and diagnosis of CRC and the relationship between CRYAB and immune infiltration in CRC have not been investigated to date. In this study, various databases such as Gene Expression Omnibus (GEO) and The Cancer Genome Atlas (TCGA) were used to explore whether CRYAB expression level can be used as an indicator of poor prognosis and also explore the potential relation of CRYAB expression level and insufficient immune cell infiltration in CRC.

## METHODS

### Data source

The Cancer Genome Atlas (TCGA) (https://genome-cancer.ucsc.edu/) provides scholars and researchers with clinical and pathological information on 33 types of cancer. Data of colorectal cancer (CRC) patients with RNA-Seq expression and matching

clinicopathological information was obtained through the TCGA tool cancer browser. Because the database is publicly available and accessible, approval from the local ethics committee was not necessary.

## GEO database
The GEO database is a comprehensive gene expression library in the National Center for Biotechnology Information (NCBI) (https://www.ncbi.nlm.nih.gov/geo/) and is one of the largest collections of gene chips in the world.

## Immunochemistry
Fresh colorectal cancer tissues and normal tissues were collected, immediately immersed in 4% paraformaldehyde overnight, dehydrated by an ethanol series (70%, 80%, 90%, and 100%), clarified in xylene, and paraffin-embedded. They were subjected to a temperature of 60 °C for 2 h, deparaffinized, hydrated with xylene and ethanol, and then the recovered nuclear antigen was washed with PBS and hydrogen peroxide solution. Additional slices were randomly selected and incubated with rabbit anti-CRYAB antibody (#15808-1-AP; Proteintech, Wuhan, China) at 4 °C overnight, followed by HRP incubated goat anti-rabbit mouse universal antibody (1: 3000, K5007, DAKO, Denmark) for 60 min at room temperature. After the slides were placed in PBS and decolorized, they were developed with DAB chromogenic solution. In addition, all slides (nuclei) were counterstained with 5 μg/mL Harris at room temperature for 3 min. Finally, the image was captured under a microscope (Nikon Eclipse E200; Tokyo, Japan).

## Statistical analysis of survival
Patients in the experimental group and the validation groups were divided into two subgroups according to the median expression of the CRYAB gene: a high CRYAB expression group and a low CRYAB expression group. The effect of CRYAB expression level on the clinical outcomes of CRC patients was investigated using Kaplan–Meier (KM) survival curves, and a prognostic classifier was constructed to compare survival differences. The KM survival curve was implemented using the "survminer" package for analysis and visualization.

## Analysis of CRYAB expression levels in colorectal cancer and normal colorectal samples
To compare CRYAB expression patterns between tumor and normal tissues, the differential expression of CRYAB was examined in TCGA datasets using the Tumor Immunity Estimation Resource (TIMER) 2.0, a comprehensive online resource for the analysis of immune infiltrates and gene expression in different cancer types (*Li et al., 2017a*; *Li et al., 2017b*; *Li et al., 2020*). In TIMER2.0, we selected "Exploration," then "Gene_De," followed by entering the gene name "CRYAB" in the text box and clicking "submit." Oncomine is a publicly accessible cancer gene chip database and web-based data mining platform containing 715 data sets and 86,733 samples (*Rhodes et al., 2004*; *Rhodes et al., 2007*). We searched the Oncomine server for human CRC, and chose the differential gene analysis segment (Normal Analysis *vs.* Cancer) to retrieve the results.

### TIMER database analysis

The association between the expression of CRYAB and the presence of five infiltrating immune cells (neutrophils, macrophages, CD4 + T cells, CD8 + T cells) and three immune-related genes (CD2, DC3D and CD3E) in CRC patients was evaluated using the TIMER database (http://timer.cistrome.org/).

### Meta-analysis

A meta-analysis of data from the TCGA, GEO, and ICGC databases was performed to evaluate the significance of CRYAB expression in CRC prognosis. Heterogeneity among the included studies was determined by the $I^2$-value obtained from the Cochrane $Q$ test and the $P$-value obtained from the chi-square test. In cases of heterogeneity ($I^2 \geq 50\%$ or $P < 0.05$), the results were summarized using a random-effects model. Otherwise, a fixed-effect model was used for analysis. The "meta" R package (R version 4.0.0) was used to perform the meta-analysis.

### GO and KEGG enrichment analyses

GO and KEGG enrichment analyses were performed using R 4.0.2 and the R packages "org. Hs.eg.db," "ggplot2," "Cluster Profiler," and "enrich plot." Only terms with $P$-value < 0.05 were considered significantly enriched.

## RESULTS

### Patient characteristics

The RNA sequencing data of 306 samples from the TCGA database and detailed clinical prognostic information were included in the analysis. Patients were divided into a low expression group ($n = 153$) and a high expression group ($n = 153$). Age, gender, and tumor size did not differ significantly between the high and low expression groups ($P = 0.1695, 0.6467, 0.3418$), however rates of lymphatic metastasis, metastasis and stage were significantly different between the high and low expression groups ($P = 8.00E{-}04$, $0.0372, 0.0101$) (Table 1).

### CRYAB expression is higher in normal tissues than in tumor samples

The CRYAB gene is significantly downregulated in CRC (Fig. 1A). TIMER analysis of CRYAB expression in various cancer types showed that CRYAB expression was significantly lower in CRC tumor tissues than in normal tissues (Fig. 1B).

An analysis of the mRNA expression levels of CRYAB in TCGA samples showed that CRYAB expression was lower in tumor samples than in normal tissues ($P < 2.22e{-}16$) (Fig. 1C). This was verified in the GEO database ($P = 0.045$) (Fig. 1F). A meta-analysis of the above three datasets was performed to evaluate the correlation between overall survival (OS) and CRYAB gene expression and to obtain more objective conclusions. Because there was no statistically significant difference between the three datasets ($P = 0.55$, $I^2 = 0\%$), a fixed effects model was used to evaluate the combined hazard ratio (HR) and 95% confidence interval (CI). A relatively high expression of the CRYAB gene was significantly correlated with poor OS (HR = 1.23, 95% CI [1.11–1.35], $P < 0.0001$; Fig. 1G), indicating that CRYAB may be a predictor of poor OS.

**Table 1 Correlation between CRYAB expression in tumor and clinicopathological characteristics of patients with colorectal neoplasms.**

| Covariates | Type | Total | High | Low | P value |
| --- | --- | --- | --- | --- | --- |
| Age | ≤65 | 145 (47.39%) | 79 (51.63%) | 66 (43.14%) | 0.1695 |
| Age | >65 | 161 (52.61%) | 74 (48.37%) | 87 (56.86%) | |
| M | M0 | 206 (67.32%) | 101 (66.01%) | 105 (68.63%) | 0.0372 |
| M | M1 | 41 (13.4%) | 28 (18.3%) | 13 (8.5%) | |
| M | unknow | 59 (19.28%) | 24 (15.69%) | 35 (22.88%) | |
| N | N0 | 180 (58.82%) | 74 (48.37%) | 106 (69.28%) | 8.00E−04 |
| N | N1 | 75 (24.51%) | 49 (32.03%) | 26 (16.99%) | |
| N | N2 | 51 (16.67%) | 30 (19.61%) | 21 (13.73%) | |
| T | T1 | 7 (2.29%) | 3 (1.96%) | 4 (2.61%) | 0.3418 |
| T | T2 | 46 (15.03%) | 18 (11.76%) | 28 (18.3%) | |
| T | T3 | 210 (68.63%) | 108 (70.59%) | 102 (66.67%) | |
| T | T4 | 42 (13.73%) | 24 (15.69%) | 18 (11.76%) | |
| T | unknow | 1 (0.33%) | – | 1 (0.65%) | |
| Gender | female | 143 (46.73%) | 69 (45.1%) | 74 (48.37%) | 0.6467 |
| Gender | male | 163 (53.27%) | 84 (54.9%) | 79 (51.63%) | |
| Stage | Stage I | 47 (15.36%) | 18 (11.76%) | 29 (18.95%) | 0.0101 |
| Stage | Stage II | 120 (39.22%) | 53 (34.64%) | 67 (43.79%) | |
| Stage | Stage III | 88 (28.76%) | 50 (32.68%) | 38 (24.84%) | |
| Stage | Stage IV | 41 (13.4%) | 28 (18.3%) | 13 (8.5%) | |
| Stage | unknow | 10 (3.27%) | 4 (2.61%) | 6 (3.92%) | |
| Expression | High | 153 (50%) | 153 (100%) | – | 0 |
| Expression | Low | 153 (50%) | – | 153 (100%) | |
| Methylation | High | 153 (50%) | 52 (33.99%) | 101 (66.01%) | 0 |
| Methylation | Low | 153 (50%) | 101 (66.01%) | 52 (33.99%) | |

**Notes.**
Abbreviations: T, Tumor; N, Node; M, Metastasis.

## Higher CRYAB mRNA expression in CRC is associated with shorter OS

According to the KM chart, CRC cases with higher CRYAB mRNA expression had a shorter OS ($P = 0.027$) (Fig. 1D) and progression-free survival ($P = 0.027$) in the test cohort (Fig. 1E). The results showed that the expression of CRYAB was lower in colorectal tumors than in normal colorectal tissues.

## Colorectal cancer CRYAB expression analysis

The expression of CRYAB in CRC was confirmed by immunohistochemistry (Figs. 2A–2F). The results confirmed that CRYAB expression was lower in CRC tissues than in normal colorectal tissues. We cut three sets of immunohistochemical slices, measured the positive rate of CRYAB among them, and drew the corresponding histogram. the positive rate of CRYAB was higher in tissue adjacent to tumors than in actual cancer tissue ($p < 0.0001$) (Fig. 2G)

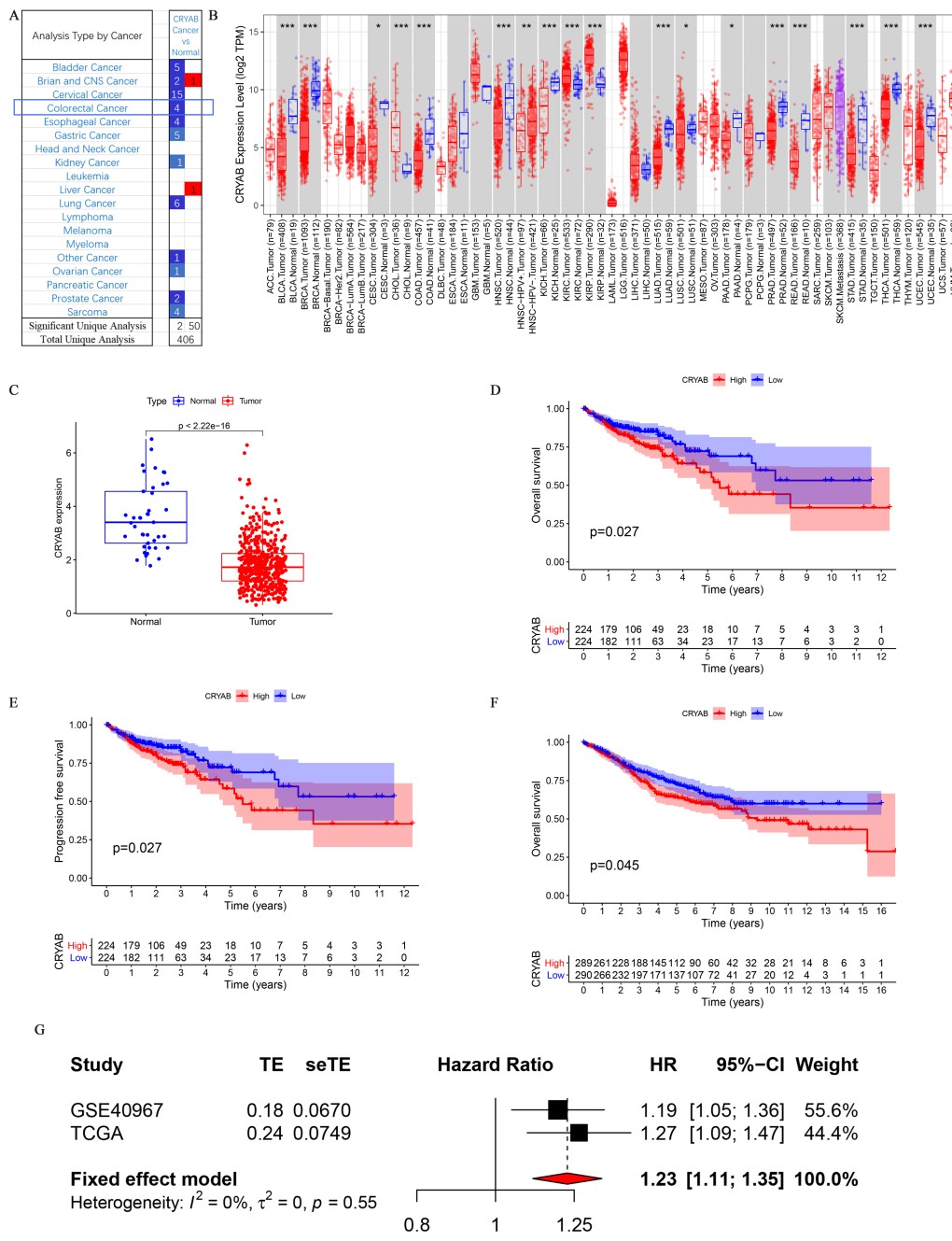

**Figure 1 Differential expression and survival analysis of the CRYAB gene.** (A) CRYAB mRNA expression in different cancers; red and blue represent downregulation and upregulation, respectively. (B) Comparative expression of CRYAB mRNA between colorectal tumor tissues and normal tissues; red and blue represent tumor and normal, respectively (statistical significance calculated by difference analysis, $^*P < 0.05$; $^{**}P < 0.01$; $^{***}P < 0.001$). (C) Human CRYAB expression levels in different cancer tissues and corresponding normal tissues. (D–F) The Kaplan–Meier survival curves of CRC patients with high and low CRYAB expression levels. (G) A meta-analysis with two data sets. High expression of CRYAB is significantly associated with poor OS. TE: estimated treatment effect; seTE: standard error of treatment estimate.

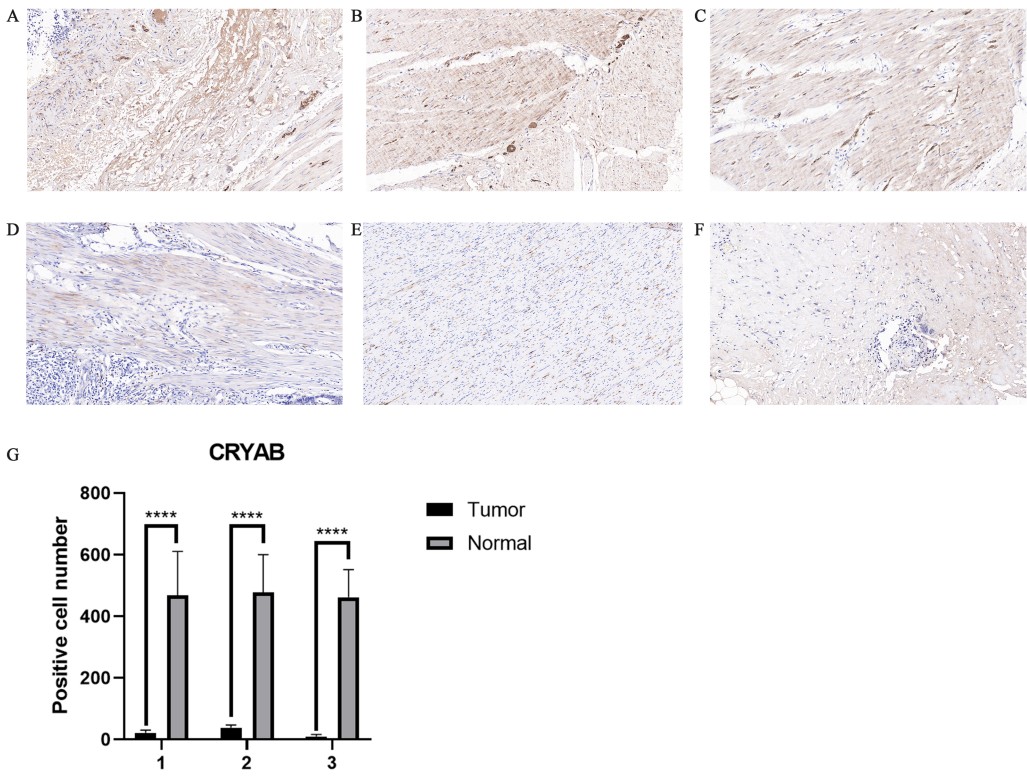

**Figure 2** **Representative IHC staining of CRYAB expression in colorectal cancer tissues and normal colorectal tissues.** Representative histopathological sections of (A–C) normal colorectal tissues and (D–F) colorectal cancer tissues stained with IHC. The colon section was incubated with HRP-labeled goat anti-pika universal antibody and stained with DAB (brown). Images are shown at 200 × magnification (scale bar, 100 μm). IHC: immunohistochemistry. (G) We cut three sets of immunohistochemical sections, including three normal tissues and three cancer tissues, and tested the number of positive cells.

## Correlation between CRYAB expression and clinical characteristics

We analyzed the association between CRYAB mRNA expression and clinicopathological parameters in CRC patients and found that CRYAB level was not correlated with gender ($P = 0.63$) (Fig. 3B). We also found that CRYAB expression decreased with age ($P = 0.026$) (Fig. 3A), and that increased CRYAB expression was associated with distant metastasis from CRC (Fig. 3C). Low expression of CRYAB was associated with colorectal cancer progression from N0 to N1 and from N0 to N2 ($P < 0.001$, $0.01$) (Fig. 3D). Low expression of CRYAB was also associated with CRC tumor size progression from T2 to T3, with an even stronger association of low CRYAB expression with the progression from T2 to T4 ($P = 0.07$, $0.02$). In terms of lymph node metastasis, the expression level of CRYAB did not change after N1 (Fig. 3E). High CRYAB expression was associated with tumor progression from Stage I to III, from Stage I to IV, from Stage II to II, and from Stage II to IV ($P < 0.001$, $0.001$, $0.05$, $0.01$). The level of CRYAB gene methylation was not associated with age, sex, TNM stage, or tumor stage (Figs. 3G–3L).
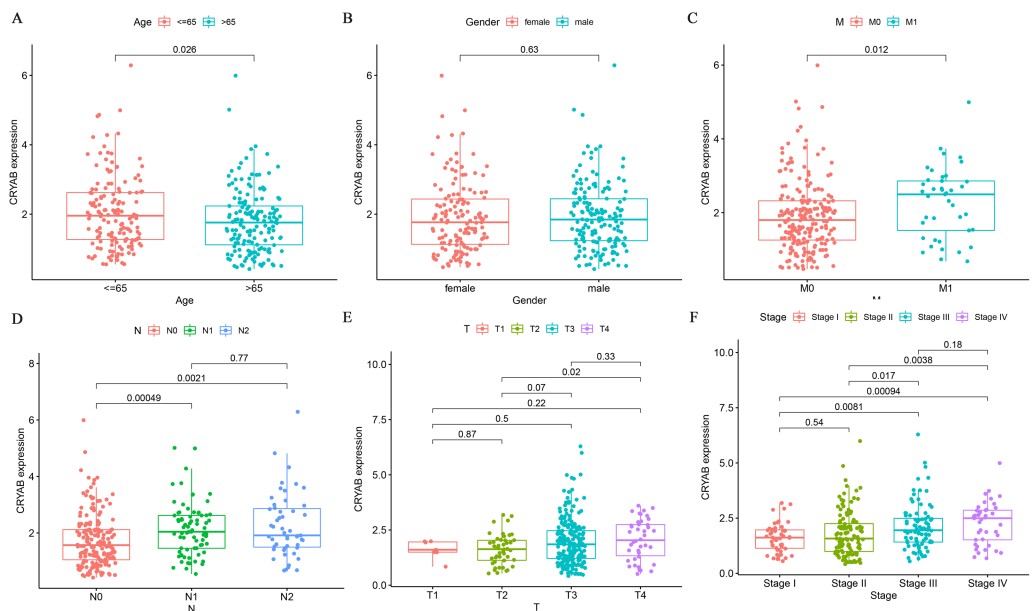

**Figure 3** **Relationship between the expression of CRYAB and clinicopathological characteristics of colorectal cancer patients.** (A–F) Box plot showing the relation of CRYAB mRNA to (A) age, (B) gender, (C) distant metastasis, (D) lymph node metastasis, (E) tumor size, and (F) cancer stage.

## Correlation analysis between infiltrating immune cells and CRYAB expression

Tumor infiltrating lymphocytes impact cancer survival (*Mou et al., 2021*). Therefore, we analyzed the correlation between CRYAB expression and four infiltrating immune cells (neutrophils, macrophages, CD8 + T cells, and CD4 + T cells) and three immune-related genes (CD2, CD3D, and CD3E). The results showed that the expression level of CRYAB is comparable to neutrophils ($r = 0.364$, $P = 4.6e−10$), macrophages ($r = 0.515$, $P = 4.23e−20$), CD4+ T cells ($r = 0.321$, $P = 4.99e−08$), CD8 + T cells ($r = 0.134$, $P = 2.6e−02$), CD2 ($r = 0.176$, $P = 1.49e−04$), CD3D (r0.176, $r = P = 1.12e−02$) and CD3E ($r = 0.209$, $P = 6.45e−06$) (2021.5.9) and that infiltration levels were significantly positively correlated with CRYAB expression with significance defined as $P < 0.05$ (Fig. 4). In addition, CRYAB expression was significantly correlated with markers of M2-like macrophages in pan-carcinoma and CRC, including TGFB1 (Fig. 5A and 5B), MRC1 (Fig. 5C and 5D), and CD163 (Fig. 5E and 5F). These results indicate that in addition to the immunosuppressive microenvironment of colorectal cancer, high CRYAB expression was related to macrophage infiltration and polarization.

## DNA methylation analysis

The degree of methylation of cg12598198 was the highest, followed from high to low by cg14276286, cg15227610, cg1158277, cg15204861, cg12947833, cg15318568, cg00514609, cg07476508, cg10048349, cg13210534, cg15545878, and cg13084335 (Fig. 6A). Among them, methylation sites such as cg13084335, cg15545878, cg13210534, and cg15318568 were positively correlated with low expression of CRYAB. The sites cg12598198,

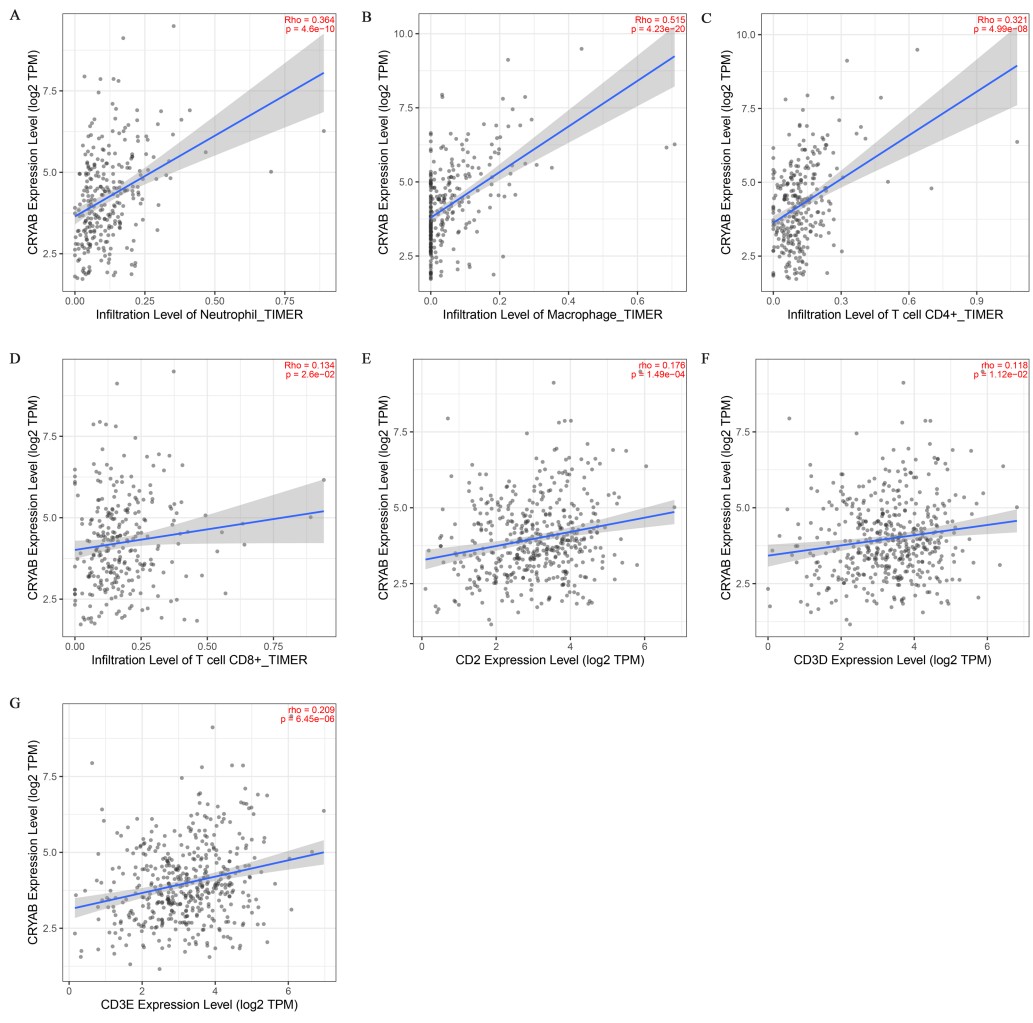

**Figure 4** **Correlation between CRYAB expression and immune infiltration in CRC.** Correlations between CRYAB expression and different immune cells: (A) neutrophils, (B) macrophages, (C) CD4+T cells, and (D) CD8+T cells. Correlations between CRYAB expression and different immune-related genes: (E) CD2, (F) CD2D, and (G) CD2E.

cg14276286, cg15227610, cg1158277, cg15204861, cg12947833, cg00514609, cg07476508, and cg10048349 and other methylation sites were negatively correlated with low expression of CRYAB (Figs. 6B–6O). The methylation level of CRYAB was lower in CRC than in the normal control group, indicating that changes in methylation are related to the abnormal expression of CRYAB. In addition, DNA methylation may be related to the molecular mechanism underlying the low expression level of CRYAB in tumor tissues and also to the pathogenesis of CRC.

## GO and KEGG enrichment analyses

We performed a GO enrichment analysis and a KEGG pathway analysis. Fig. 3A shows the top 30 significantly enriched upregulated pathways. The GO pathway analysis showed that relatively high expression of CRYAB was associated with extracellular

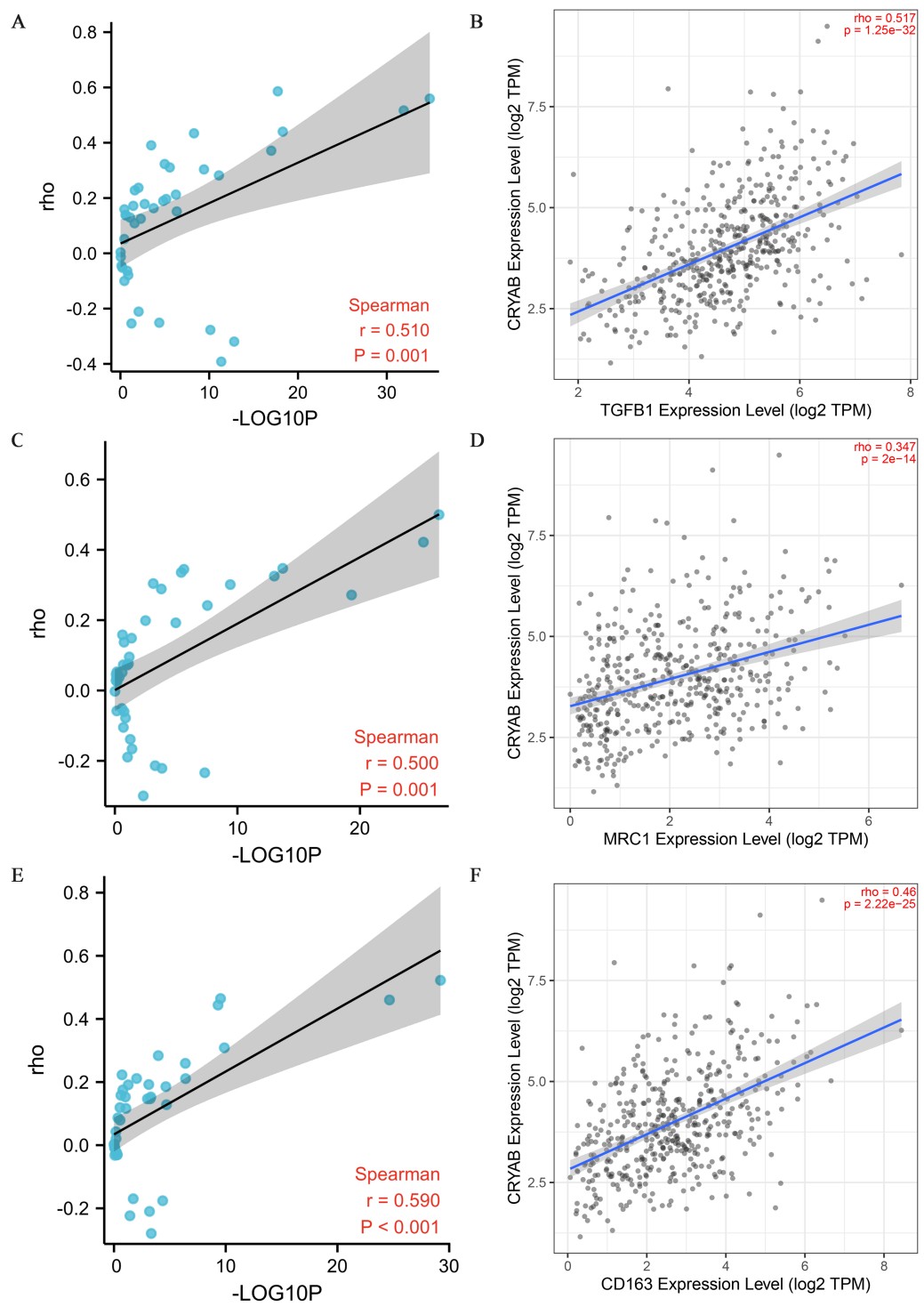

**Figure 5 Correlation between CRYAB expression and genetic markers of M2-like macrophages.** (A, B) Correlation between CRYAB expression and TGFB1 in pan-carcinoma (A) and CRC (B). (C, D) Correlation between CRYAB expression and MRC1 in pan-carcinoma (C) and CRC (D). (E, F) Correlation between CRYAB expression and CD163 in pan-carcinoma (E) and CRC (F). $P < 0.05$ indicates statistical significance.

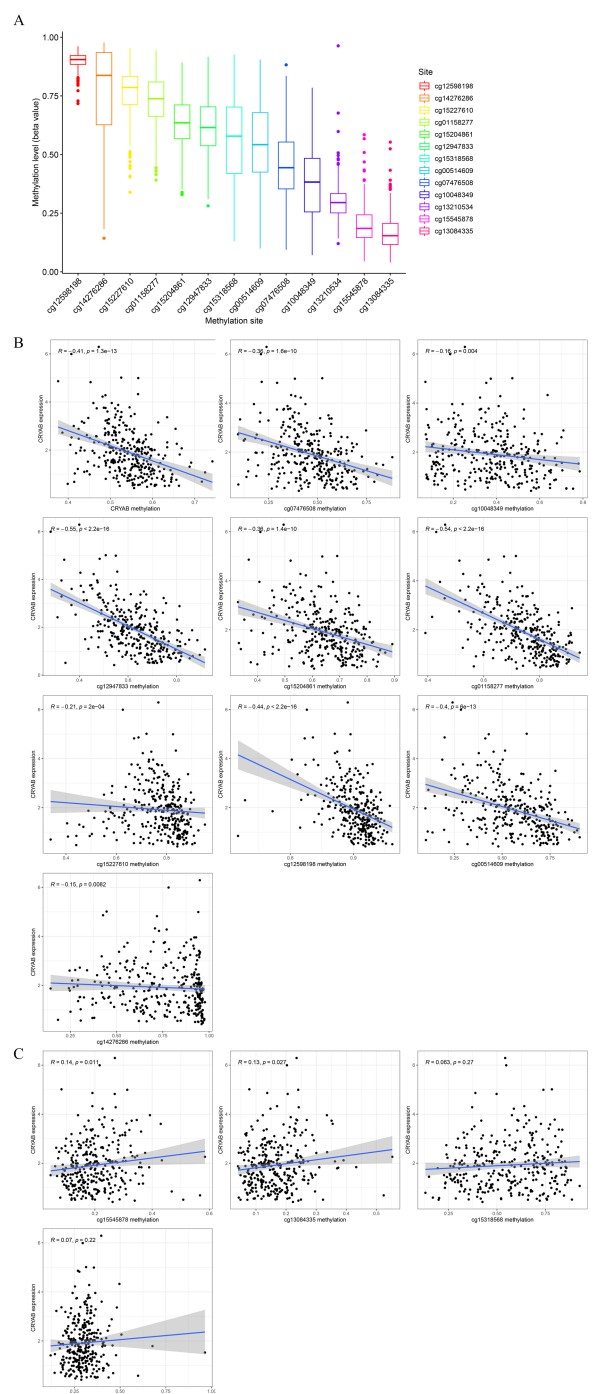

**Figure 6 Relationship between CRYAB methylation and CRYAB gene expression in CRC.** (A) Methylation level of each methylation site. (B–C) Correlation between CRYAB gene methylation site and CRYAB gene expression.

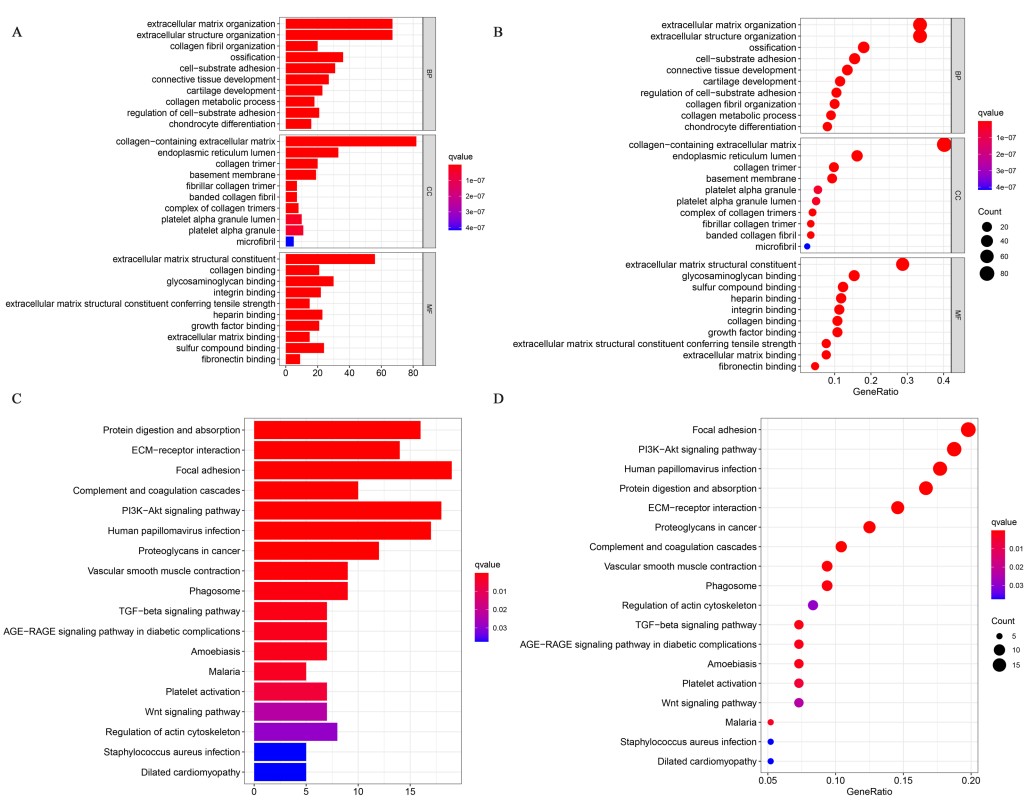

**Figure 7 GO and KEGG enrichment analyses.** (A) Gene enrichment in three different GO functions and (B) KEGG pathways were respectively ranked by *p*-value and gene enrichment count.

structure organization (GO:0043062, $P = 2.13E−64$), ECM organization (GO:0030198, $P = 1.76E−64$), collagen-containing ECM (GO:0062023, $P = 3.59E−85$), and ECM structural constituent (GO:0005201, $P = 3.86E−70$) (Fig. 7A). The KEGG pathway analysis identified many enriched pathways, including PI3K-Akt signaling pathway, focal adhesion, human papillomavirus infection, ECM-receptor interaction, and protein digestion and absorption, and that CRYAB genes were strongly linked to them (Fig. 7B)

# DISCUSSION

CRYAB is a principal member of the small molecule heat shock protein family (*Annertz et al., 2014*). CRYAB functions primarily as a molecular chaperone, preventing other proteins from stress injury, including those caused by heat shock, radiation, and oxidative stress (*Moyano et al., 2006*). Research shows that CRYAB can promote the occurrence and development of tumors (*Li et al., 2017a*; *Li et al., 2017b*; *Shi et al., 2017*). M2 macrophages promote NSCLC metastasis by upregulating CRYAB (*Guo et al., 2019*). We found that CRYAB is downregulated in CRC, which is contrary to the results of previous studies (*Shi et al., 2014*). The gene expression in different tumors was inconsistent, which may be related to the specific environment (*Dey et al., 2021*), because gene expression is affected by many factors, such as the expression of PDCD1 in different tumors. PDCD1 expression

is induced by TCR and/or B-cell receptor signaling, but this expression is frequently enhanced by tumor necrosis factor (TNF) stimulation (*Nakae et al., 2006*). The underlying mechanism may involve different patterns of CRYAB phosphorylation in different tumors, which determine the protein-binding library and biological effects of each tumor (*Kuipers et al., 2017*).

We identified CRYAB as a new potential therapeutic target and predictive biomarker for CRC and found that CRYAB can be used as a prognostic indicator of immune status. We also found that relatively high expression of CRYAB and the methylation of CRYAB in tumor tissues are both related to the infiltration of immune cells. The methylation level of CRYAB was not related to age, gender, TNM stage, or tumor stage in our analyses.

The KEGG pathway analysis and GO functional enrichment analysis showed that CRYAB was significantly associated with tumorigenesis and tumor development pathways, including: focal adhesion for cell migration (*Paluch, Aspalter & Sixt, 2016*), the PI3K-AKT signaling pathway, ECM organization, human papillomavirus infection leading to cervical cancer (*Schiffman et al., 2016*), and ECM-receptor interaction leading to melanoma metastasis (*Chen et al., 2019*). The intestinal ECM is mainly composed of collagen, which is essential for regulating cell division, differentiation, proliferation, growth, migration, and apoptosis indicating that it plays a vital part in the development and progression of cancer (*Fischer et al., 2001*). Therefore, this indicates that the high expression of CRYAB is involved in the positive regulation of these signal pathways. CRYAB may play a pathological role in promoting tumor cell proliferation by driving the up-regulation of these signal transduction pathways. In addition, CRYAB may inhibit tumor cell migration and invasion through overexpression in T24 and J82 BC cell lines (*Ruan et al., 2020*). The high expression of CRYAB could promote the proliferation, invasion, and metastasis of CRC through EMT (*Zhang et al., 2019a*; *Zhang et al., 2019b*). Its expression level in CRC patients is closely related to the two core EMT gene products, MMP7 and E-cadherin. Furthermore, three important signaling pathways (PI3K, p38, and ERK) are involved in CRYAB-induced EMT (*Worthley & Leggett, 2010*; *Giordano et al., 2015*; *Li et al., 2017a*; *Li et al., 2017b*). Hence, although CRYAB can be used as a tumor-suppressor gene in CRC, the expression of CRYAB in highly malignant cancers is increased. We used the TIMER database to reveal, for the first time, that the expression of CRYAB in CRC is associated with the infiltration of a variety of immune cells. Tumor-infiltrating lymphocytes, such as tumor correlated macrophages and cancer-infiltrating neutrophils, may affect the prognosis and efficacy of chemotherapy and immunotherapy (*Waniczek et al., 2017*; *Zhang et al., 2018*). The level of tumor infiltration by immune cells is correlated with tumor growth, progression, and patient outcome (*Gajewski, Schreiber & Fu, 2013*). A close relationship between immune infiltration and the occurrence and development of CRC was reported previously (*Xiong et al., 2018*), however, there were no studies analyzing the relationship between CRYAB expression and immune cell infiltration. We evaluated the relationship between CRYAB expression and the immune infiltration level of CRC using the TIMER website. CD2, CD3D, and CD3E genes were positively related to the infiltration of neutrophils, macrophages, CD8 + T cells, CD4 + T cells, and CD2, CD3D, and CD3E genes. Our findings confirm that the increased infiltration levels of immune cells are crucial for the progression of CRC,

and CRYAB expression is a predictor of that increase of immune cell infiltration. These results indicate the need for further research on the relationship between CRYAB and CRC immune infiltration.

The present study had several limitations. We only obtained data from TCGA, including mRNA expression and methylated expression, which may lead to data deviation in this study. A larger number of tumor specimens and further experimental verification are needed to evaluate the biological role of CRYAB in CRC. In addition, further experimental validation is needed to determine the value of CRYAB for predicting the prognosis of CRC and for use in the formulation of treatment strategies.

## CONCLUSION

The present study provides important evidence supporting the importance of CRYAB in the prognosis of human CRC. We identified CRYAB as a novel biomarker and clarified its prognostic potential in CRC through the analysis of online public databases. Its biological function and role in immune infiltration were examined to elucidate the mechanism underlying its relatively high expression, which is the basis of CRC. Our results indicate that CRYAB can be used as a potential tumor suppressor gene in CRC. CRYAB may also be a new potential therapeutic target and predictive biomarker for CRC. Our experimental data provides insights that can be used for the development of appropriate treatment strategies and to further research on the topic. In addition, we are committed to studying cancer cell lines and mouse models of CRC to validate the present findings and develop effective treatment strategies by targeting the CRYAB gene.

### Funding
This study was funded by the Department of Finance of Hubei Province and the Department of Science and Technology of Hubei Province, under a special project of the Hubei Provincial Central Government for guiding local science and technology development (Xiaodong Huang, grant no. 2019ZYYD067), and by the Youth Project of Health Commission of Wuhan (Xiaoli Chen, grant no. WX20Q17). The funders had no role in study design, data collection and analysis, decision to publish, or preparation of the manuscript.

### Grant Disclosures
The following grant information was disclosed by the authors:
Department of Finance of Hubei Province and the Department of Science and Technology of Hubei Province: 2019ZYYD067.
Youth Project of Health Commission of Wuhan: WX20Q17.

### Competing Interests
The authors declare there are no competing interests.

## Author Contributions

- Junsheng Deng conceived and designed the experiments, performed the experiments, analyzed the data, prepared figures and/or tables, and approved the final draft.
- Xiaoli Chen conceived and designed the experiments, performed the experiments, authored or reviewed drafts of the paper, and approved the final draft.
- Ting Zhan, Mengge Chen and Xisheng Yan analyzed the data, prepared figures and/or tables, and approved the final draft.
- Xiaodong Huang conceived and designed the experiments, authored or reviewed drafts of the paper, and approved the final draft.

## Data Availability

The data is available at UCSC Xena and NCBI GEO https://www.ncbi.nlm.nih.gov/geo/query/acc.cgi?acc=GSE40967 and GPL570, Oncomine, TIMER2.0.

Figure 1A can be found at Oncomine: https://www.oncomine.org/resource/main.html.

Enter CRYAB in the search box, then select Primary Filters—Differential Analysis—Cancer vs Normal analysis, in the Dataset Filters—Data Type—mRNA below, in Other Views, select Gene Summary View, you can get a corresponding picture.

Figures 1B and 4 can be found at TIMER2.0: Available at http://timer.cistrome.org/.

Figure 1C can be found at Xena: http://xena.ucsc.edu/, click "Launch Xena", enter "DATA SETS", click "GDC TCGA Colon Cancer (COAD)", enter "HTSeq-FPKM ($n = 512$) GDC Hub" of "gene expression RNAseq".

Figure 1D can be found at Xena: Available at http://xena.ucsc.edu/, click "Launch Xena", enter "DATA SETS", click "GDC TCGA Colon Cancer (COAD)", enter "survival data ($n = 539$) GDC Hub" of "phenotype".

Figure 1E can be found at Xena: Available at http://xena.ucsc.edu/, click "Launch Xena", enter "DATA SETS", click "TCGA Pan-Cancer (PANCAN) (41 datasets)", Enter "Curated clinical data (n=12,591) Pan-Cancer Atlas Hub" of "phenotype".

Figures 3 and Fig. S1 can be found at Xena: http://xena.ucsc.edu/, click "Launch Xena", enter "DATA SETS", click "GDC TCGA Colon Cancer (COAD)", enter "Phenotype ($n = 571$) GDC Hub" of "phenotype".

Figure 6 can be found at Xena: http://xena.ucsc.edu/, click "Launch Xena", enter "ATA SETS", click "GDC TCGA Colon Cancer (COAD)", enter "Illumina Human Methylation 450 ($n = 347$) GDC Hub" of "DNA methylation".

https://www.ncbi.nlm.nih.gov/geo/query/acc.cgi?acc=GSE40967.

https://www.ncbi.nlm.nih.gov/geo/query/acc.cgi?acc=GPL570.

https://ftp.ncbi.nlm.nih.gov/geo/series/GSE40nnn/GSE40967/matrix/.

## Supplemental Information

Supplemental information for this article can be found online at http://dx.doi.org/10.7717/peerj.12578#supplemental-information.

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
