# Peer review of "CRYAB predicts clinical prognosis and is associated with immunocyte infiltration in colorectal cancer"

_PeerJ, doi:10.7717/peerj.12578_

## Round 0.1 · original submission · Major Revisions

Thank you for submitting your manuscript to PeerJ. The two reviewers of your paper were positive about its contributions but noted some concerns and made suggestions that I invite you to address.

Reviewer 1 suggests more detail be provided about how data were accessed and how analyses were conducted, especially the GO and KEGG analyses. They also suggested added detail for the figure legends and provide specific suggestions for figure edits. Their biggest concern were statements making a causal link between CRYAB expression and colon cancer that you may want to address.

Reviewer 2 made suggestions for how to clarify some information throughout the paper to be more specific about the approach and interpretation of data. They had a question about the interpretation of immunohistochemistry results in figure 2 that should be addressed. They also recommended an overall review of the paper for grammatical errors.

I invite you to submit a revised manuscript after major revisions and ask that your rebuttal letter address all reviewer comments.

I look forward to receiving your revised submission.

Reviewer 1 ·

Basic reporting

The authors have analyzed data from multiple publicly available databases to investigate the role of CRYAB in human colorectal cancer, showing that while CRYAB expression is lower overall in CRC cells, relative CRYAB mRNA levels are inversely correlated with survival and that CRYAB expression positively correlates with immunocyte infiltration. The authors implicate methylation status of specific gene body loci as potential regulators of CRYAB expression levels. This work provides a potential for CRYAB expression levels to serve as predictive indicators of prognosis for individuals with colorectal cancers.

The paper would be improved with a more thorough discussion of CRYAB expression in colorectal cancer and perhaps in cancer in general- there is some information about CRYAB expression in multiple cancers in Figure1B, but especially with this included, a more thorough background in the text would be useful. It also would be worthwhile to discuss some of the work done by the Pingsheng Chen lab on CRYAB expression in CRC.
The writing throughout the paper also needs to be improved for clarity. In several sections there was some unclear or contradictory language such as in lines 112-113, 130-131, 168, and there is also a sentence which appears to belong in a paper investigating Girdin expression in hepatocellular carcinoma (line 96).

The paper would also be improved with more thorough and descriptive figure legends.

Experimental design

Overall, the manuscript would benefit from far more detailed information regarding how the data presented were obtained and analyzed- without that information it is difficult to adequately evaluate the experimental design. Additionally, there should be a date of access included for each piece of data. When replicating some of the graphs presented in the paper, for example, I found that for Figure 4, there were several minor differences in Rho and p values (there were no differences for Fig 4 F-G) when I accessed the TIMER database, so those should be updated in the manuscript. The authors should also clearly note that these data AND the statistical calculations are from TIMER, and also do so where appropriate for other analyses performed. Of particular importance, in my opinion, is the addition of information regarding exactly how Gene Ontology and KEGG analyses were performed. This may be easily inferred by a reader who has experience with Go and KEGG, but as someone who has not used these tools, I would have liked more detailed information.

Validity of the findings

My biggest concern with the paper is that in many cases correlation is used to make claims about causality that could only be determined with targeted work to test these hypotheses.

In many cases the authors conclude that CRYAB is exerting a biological effect based on simple correlation. For example, the authors state that, “expression of CRYAB *leads to* focal adhesion, PI3K-Akt signaling pathway, Human papillomavirus infection…. (line 181), which cannot be concluded from this analysis. The correlation of CRYAB expression levels and immune response in a complex biological environment is not sufficient to claim that CRYAB “plays a key role in regulating tumor immunology (lines 260-261)” or that “the relative high expression of CRYAB may be a potential mechanism of *CRYAB induced* colon cancer (lines 257-258)”. This is an interesting hypothesis worthy of further investigation, but the data presented here shows only a correlation and is not sufficient to draw these conclusions without some targeted work testing these hypotheses regarding CRYABs role in CRC.

I do agree with the authors that CRYAB expression levels could be useful in a clinical setting to potentially function as a prognostic indicator in CRC patients, and this work lays some groundwork for informing experiments to test how CRYAB may function on a molecular level in colorectal cancers. With revisions to some of the interpretation of these findings, this paper presents sufficient data for a publication; however, some further work investigating some of those claims would dramatically improve the paper’s impact.

Additional comments

Below are some comments on specific figures and how they may be improved:

Figure 1:
It is not clear why the authors chose to include figures (A and B) showing CRYAB expression levels in a variety of cancers rather than just COAD, since the paper does not address CRYAB expression levels in these other cancer types (many of which showing a significant change in expression levels compared to wild type). Previous published work has also shown an increase in CRYAB expression levels in COAD cancer cells (Shi et al 2014) and this should be addressed. Figure 1 could be improved by changing the color scheme for either A or B, since red and blue correspond to expression level in A but not in B. Crucial information regarding how TCGA was used to generate these plots is missing and I could not easily find on the TCGA web portal how one might replicate these.

Figure 2:
1. The exact source of these images is needed, including not just the data repository but also patient/sample number. Additionally, the antibody used should be reported.
2. The images themselves do not with any degree of certainty demonstrate that there is a difference in CRYAB protein levels between normal and cancerous tissue. This is further compounded by the issue of different cell morphology in cancerous tissue, making comparison difficult. While it is understood that the authors are limited by the data available in the Human Protein Atlas Project, without the ability to distinguish a change in staining levels within tissue from the same patient, these data are not conclusive.

Figure 3:
The plots would be improved with clear/more descriptive legends for each as well as an n value for each measurement. For the methylation data, none of these reach the threshold of significance; this would be more appropriately included in supplemental data and not the body of the paper.

Figure 5:
The legend indicates that purple circles represent meaningful correlations, however both on my monitor and printed, none of the circles are purple.

Figure 6:
It would be useful if these were grouped such that methylation loci correlating with increased CRYAB expression are together, and vice versa.

Reviewer 2 ·

Basic reporting

Abstract:

1. Conclusion.CRYAB may be a potential tumor suppressor
gene of COAD. CRYAB may be a new potential therapeutic target and predictive biomarker
for CRC. CRYB likely plays an important role in immune cell infiltration.

a. Please correct CRYB to CRYBA to be consistent.
b. What does the abbreviation CRC mean? Is it the same as COAD?

Introduction:

1. Add prevalence of the disease (how many people or percent of population affected in U.S.?)

2. Line 49 domain, and a C-terminal domain[11]. The study report that CRYAB may trigger the EMT in
50 CRC by activating the ERK signaling pathway and is a potential tumor biomarker for CRC
51 diagnosis and prognosis[12].

Please expand on the rationale for studying infiltration of immune cells in COAD. Has immune cell infiltration been attributed to worse tumor progression?

Experimental design

Methods

1. Please provide the methods for the immunohistochemistry that is presented in Fig 2.

Validity of the findings

Results:

1. Line 103 In general, the RNA sequencing data of 306 samples from the TCGA database and detailed
104 clinical prognostic information resources have been included in our study.

What is meant by “in general”?

2. Line 104 clinical prognostic information resources have been included in our study. All patients were
105 randomly divided into low expression group (n =153) and high expression group (n =153).

What is meant by “randomly divided”. Is the dividing into low and expression groups is based on the data of expression levels for CRYAB? Were the levels rank ordered and then divided at the midline so as to get exactly 153 in “low group” and 153 in “high group”? If so, this does not sound like a scientific approach. Please provide details on methodology.

3. Line 107 (p=0.1695, 0.3418, 0.6467). M0, N0, and Stage I are different

What is meant by “different”? Please expand.

4. Line 111 cancer (Figure.1A). TIMER analysis revealed the comparison of CRYAB gene among various
112 cancer types including colon cancer, and indicated that CRYAB of colon adenocarcinoma
113 Significantly down-regulated genes(Figure.1B).

Correct to “is significantly” ?

What genes are "down-regulated" exactly ? Please provide a line or two about this.

5. Table 1: Is colorectal adenocarcinoma the same as colorectal neoplasm? Text talks about adenocarcinoma but table lists neoplasms.

Please be precise in table legend and write the location of the CRYAB expression - presumably in the tumor ?

Also add to Table 1 legend what the abbreviations mean such as the Type N1, N2, T1, T2…

6. Line 117 (Figure.1 F). We then performed a meta-analysis on the above three data sets to evaluate the
118 association between CRYAB gene expression and OS and obtain more objective conclusions.

What does the abbreviation OS mean?

7. Figure 2:

The immunohistochemistry claim that CRYAB staining is increased in COAD is very unconvincing.
Please provide inserts of larger magnification and perform quantitation.
Can you identify what cells the CRYAB is expressed and increased in?

Additional comments

1. This manuscript provides potential interesting results regarding CRYAB as a biomarker of tumor progression in COAD.
2. Please correct ‘nuetrophil’ spelling to ‘neutrophil’ thoughout the paper.
3. Manuscript could benefit from improved grammar.

---

## Round 0.2 · Minor Revisions

Thank you for submitting your revised manuscript to PeerJ. Both reviewers were positive about your edits and have only some additional minor comments for your consideration.

Most significantly, Reviewer 1 suggests some additional discussion of why CRYAB may show decreased expression in some tumors but increased expression in others. Reviewer 2 suggests some further general editing for grammar and readability along with a few minor suggestions. For example, I noticed that the first sentence of the discussion is incomplete. PeerJ offers editing services if you would find that helpful.

I invite you to submit a revised manuscript after minor revisions and ask that your rebuttal letter address all reviewer comments.

I look forward to receiving your revised submission.

Reviewer 1 ·

Basic reporting

Paper has been much improved- language is clearer, experiments and importance of research very nicely motivated in the introduction. I only have a few minor suggestions.

1. Please provide a citation for line 163, "Tumor infiltrating lymphocytes affect the survival of various cancer patients."

2. Please provide a citation for lines 213-215, "The intestinal ECM...apoptosis."

3. Please provide a brief definition for the TNM cancer staging system prior to its first use.

Experimental design

Very happy with the addition of information about the databases used and methods, as well as the changes to figures, especially regarding the work done for Fig 2.

Validity of the findings

While the decreased expression of CRYAB in COAD tumor tissue compared with normal tissue is consistent with the conclusion that CRYAB could function as a tumor suppressor gene in COAD, the increased expression of CRYAB in highly malignant cancer suggests the opposite. So some explanation or more in depth discussion of how this increased malignancy with increased CRYAB expression fits with the conclusions here would be helpful, even if it's speculation clearly stated as such.

Reviewer 2 ·

Basic reporting

General
1. The revised manuscript is improved. The manuscript could still benefit from further grammatical and readability improvements.

Abstract

1. Line 18 Methods
Please use a more scientific term than “ checked”, for example ‘examined’, ‘investigated’, ‘studied’,’assessed’, ‘quantified’

Introduction

Line 45-54
Is the revised added data about CRC prevalence worldwide or for the U.S.? Please provide in manuscript.

Experimental design

Methods

1. Line 106: . Serial sections were generated and select DAB.
Should this sentence be corrected to: Serial sections were collected?
Shouldn’t DAB stain be used after applying the primary and secondary antibodies? Please clarify methodology.

The immunohistochemistry methodology does not match what is written in Figure 2 legend. Please make it consistent.

Validity of the findings

No comment.

---

## Round 0.3 · accepted · Accept

Thank you for your consideration and response to the remaining reviewer comments. I am happy to now accept your paper for publication in PeerJ.

You will be given the option to make the reviews of your manuscript available to readers. Please consider doing so as this review record can be a great resource for readers of your paper and contributes to more transparent science.

Thank you for choosing PeerJ as a venue for publishing your work.